# Effect of Ceramic and Nylon Fiber Content on Composite Silica Sol Slurry Properties and Bending Strength of Investment Casting Shell

**DOI:** 10.3390/ma12172788

**Published:** 2019-08-30

**Authors:** Pengpeng Huang, Gang Lu, Qingsong Yan, Pu Mao

**Affiliations:** National Key Laboratory of Light Alloy Processing Science and Technology, Nanchang Hangkong University, Nanchang 330063, China

**Keywords:** ceramic fiber, nylon fiber, investment casting, silica sol ceramic shell, wet strength, high strength, movement viscosity

## Abstract

In order to improve the performance of the investment casting shell, ceramic and nylon fiber was added to the silica sol slurry to study the effect of ceramic and nylon fiber on the liquidity of the silica sol slurry and the bending strength of the investment casting shell. Meanwhile, the fracture surface of shell sample was observed by SEM. The experiment results indicate that the movement viscosity of slurry increases with ceramic and nylon fiber content, increasing from 0 to 0.75 wt.%. The movement viscosity of ceramic fiber composite slurry is higher than nylon fiber composite slurry. The wet and high strength of shell firstly increases then decreases, with an increase of ceramic and nylon fiber content. When the ceramic and nylon fibers’ contents are 0.6 wt.%, the maximum wet strengths are 3.56 MPa and 3.84 Mpa respectively, increasing by approximately 38% and 43%. Moreover, the high strength of shell achieves its highest value, 5.08 Mpa, when the ceramic fiber content is 0.6 wt.%; however, when the nylon fiber content was more than 0.3%, the high strength of the nylon fiber reinforced shell was decreased sharply. Therefore, an addition of ceramic and nylon fiber to silica sol slurry distinctly influences the wet and high strength of investment casting shell.

## 1. Introduction

Investment casting is extensively used in engineering applications, such as engine valves, vanes and turbochargers and is now being used in aero-engines [1,2,3]. This possible due to its casting’s excellent surface quality, high-dimensional accuracy and suitability for producing highly complex and near-net shaped parts [4,5,6]. The investment casting shell is considered to be an especially suitable processing method for the manufacture of many alloys, and is prepared by silica sol binder [7,8,9,10,11]. Silica sol is a kind of high-quality binder for the preparation of ceramic shell; it has such advantages as simple preparation, good slurry stability, little shell shrinkage and high surface finish castings [12,13]. However, it introduces other issues, such as low wet-strength and high residual strength. Low wet-strength could lead to shell cracking during the handling and dewaxing, and high residual strength could cause the poor quality of sand cleaning after the shell casting [14,15,16]. The integrative capability of ceramic shell is dependent on the slurry and shell materials, as well as the process by which the shells are built. Thus, improving the materials and production process of ceramic shell become very important to ensure the high quality of ceramic shell.

Fiber is a new type of lightweight material that can be used as reinforcement material by adding into the corresponding dispersion medium, to prepare high performance composite materials [17,18,19]. Up to now, domestic and foreign scholars have found that mechanical properties and microstructure of the composite matrix can be effectively improved by using fiber as a reinforcement to join metal matrix composites [20,21,22]. Researchers have added fibers as reinforcing materials to the casting slurry to prepare the investment casting shell, thereby enhancing the performance of the shell. There is some research [23,24] that ceramic shell, reinforced through the adoption of nylon fiber dispersion in a ceramic slurry for investment casting, could be effectively improved in terms of tensile performance, and resist cracking of the ceramic shell. Another significant advantage of the nylon fiber additions appears to be a greatly increased permeability after firing. An addition of ceramic fiber to the ceramic investment slurry material distinctly increases the ceramic shell strength, further inhibiting, effectively, the production of shell cracks [25,26]. The presence of the rice husk fiber has reinforced the shell mold body after the firing process by increasing the porosity and ultimately permeability of the ceramic shell [27].

Due to the different types of fibers’ characteristics, there are differences in the enhancement effects on investment casting shell. The paper mainly studies the effect of ceramic and nylon fibers on the liquidity of the silica sol slurry, and the bending strength of investment casting shell. Moreover, it focuses on fiber content’s influence on shell properties, and further discusses the fiber strengthening mechanism, which provides theoretical guidance for obtaining high performance investment casting shell which will effectively refrain the cracks.

## 2. Materials and Methods

The white corundum powder and mullite were adopted as refractory materials of investment casting ceramic shell; the purity of white corundum powder was 99.16%. Alkaline silica sol which contained 29–31 wt.% colloidal SiO_2_ particles was used as ceramic shell binder. The surface morphology of ceramic and nylon fiber used in this study are shown in Figure 1. As shown in Figure 1, the morphology of nylon fiber is more uniform and orderly than the ceramic fiber. The main performance indicators of fibers are shown in Table 1.

The current process was used to prepare the investment slurry of the primary and transition layers. In order to keep a good surface finish for castings during the pouring process, the ceramic and nylon fibers were only added to the slurry backup and seal layer. Firstly, the surface of fiber was modified with CG-221 type silane coupling agent, which can improve the wettability between fibers and silica sol; secondly, the fibers were added and dispersed uniformly into the silica sol. The synergistic effect of the TJS-3000 ultrasonic generator (Hangzhou Success Ultrasound Equipment Co., Ltd., Hangzhou, China) and the GZ120 electric agitator (Poly Research Equipment Co., Ltd., Jiangyin, China) was used to disperse the fiber composite slurry; finally, the white corundum powders were added into the fiber composite slurry. In order to mixed sufficiently, the speed of the electric mixer ws set to 120 r/min. The stirring time was 12 h, and the stirring temperature was set to room temperature. The change of fiber content was between 0 and 0.75 wt.%; according to the mass ratio of silica sol, an increase of each interval is 0.15 wt.%. The viscosity of slurry with ceramic and nylon fiber were measured at room temperature by LND-1 viscometer (Shanghai Base Solid Instrument Co., Ltd., Shanghai, China). The movement viscosity of fiber composite slurry was calculated as:(1)$$\upsilon =\frac{t-11}{0.154}\left(t<23\;s\right)$$
where *υ* (mm^2^/s) is the movement viscosity of the slurry and *t* (s) is the time of flowing.

For the bending strength testing, the ceramic shell samples were prepared upon a wax pattern, and the wax pattern size was 70 mm × 22.36 mm × 11.18 mm. The preparation process of the ceramic shell samples is shown as Table 2. In the process of shell making, only the big surface of wax pattern was coated; the slurry around ceramic shell sample was scraped after coating every time.

The roasting temperature of investment casting shell was 1050 °C. Firstly, temperature is risen to 450 °C, and kept for 30 min. Then, temperature is slowly risen to 1050 °C which is the roasting temperature, and kept there for 120 min; lastly, shell samples are cooled to room temperature inside the furnace. The process of shell roasting is shown as Figure 2. To evaluate the mechanical properties of the ceramic shell, three-point bend tests were conducted at room temperature. The bending strength of ceramic shell samples was tested on a XQY-II intelligent sand strength machine (Kunshan Guangmao Instrument Equipment Co., Ltd., Suzhou, China). Quanta 200 scanning electron microscopy (SEM, FEI, Hillsboro, OR, USA) was used to observe the fiber morphology and the fracture morphology of investment casting shell samples. The samples analyzed by SEM were sputtered by the conductive layer before SEM imaging. Sputtering distance was 50 mm; sputtering current was 10mA; sputtering time—50 s.

## 3. Results and Discussion

### 3.1. Effect of Fiber Content on Movement Viscosity of Silica Sol Slurry

The effect of ceramic and nylon fiber contents on movement viscosity of fiber composite silica sol slurry can be seen in Figure 3. The movement viscosity of composite silica sol slurry increases when ceramic and nylon fiber content increases. The movement viscosity of the ceramic fiber slurry was higher than nylon fiber slurry. When the ceramic fiber content was less than 0.3 wt.%, the increasing trend was stable, but when the ceramic fiber content was more than 0.3 wt.%, the growth was more obvious. For the nylon fiber slurry, the movement viscosity was close to the linear change; because the nylon fiber was relatively uniform, the surface performance was more stable, and the composite slurry performance was more stable.

Investment casting slurry is a suspension system of high solid content; silica sol as a decentralized system has certain viscosity. Every group and particle of the decentralized system may be affected by hydrogen bonding, electrostatic and van der waals force, or ionic bonding and so on, which would form the three-dimensional network structure in the quiescent state [28,29]. Because the nylon fiber has a good compatibility with colloidal silica dispersions, nylon fiber can be evenly dispersed in the spacial network structure, and combined more closely with particles in the slurry. The movement viscosity of the nylon fiber slurry is significantly lower. For the ceramic fiber slurry, the compatibility between ceramic fiber and silica sol is weaker; the dispersion of ceramic fiber is not good; the phenomenon of ceramic fiber interweaving together in the slurry appeared with an increase of fiber content. In the decentralized system, the apparent volume of powder particles and fiber has changed. Its huge network structure of fiber produced the elastic bending, which leads to the cohesion being sharply increased. It then formed a greater resistance for the flowing of fiber, so the movement viscosity of the slurry became higher. Therefore, when ceramic and nylon fiber content increases, the movement viscosity of the silica sol slurry increases respectively, but the nylon fiber slurry is relatively slower. The liquidity of nylon fiber slurry is good, so it is more conducive to coating ceramic shell.

### 3.2. Effect of Fiber Content on Green Bending Strength of Investment Casting Shell

The changing curves of the green bending strength with increasing of fiber content are shown in Figure 4. It can be seen that the green bending strength of shell first increases then decreases with an increase of ceramic and nylon fiber content. When fiber content increases from 0% to 0.6%, green bending strength of the shell increases gradually, and the increase is extremely obvious and linear. When the fiber content is 0.6%, green bending strength of shell with ceramic and nylon addition fibers is highest, at 3.56 MPa and 3.84 MPa, respectively, increases of 38% and 43% compared with the unreinforced shell. However, when fiber content continues to increase, the green bending strength begins to reduce; when the fiber content is 0.75%, the green bending strength is still higher than traditional shell. This is because that fiber disperses in shell and forms tight network framework, there fibers have strong toughness and high tension resistance, which carry a certain force to play a role of the stiffener when the ceramic shell will be fractured [30,31]. Compared with ceramic fiber, nylon fiber is more appropriate and has good compatibility with silica sol slurry. The movement viscosity of nylon fiber composite slurry is lower, which leads to the combination with the shell being more effective. The nylon fiber exists in single filaments, evenly distributed in shell. The strengthening effect of nylon fiber on the shell is more powerful as well. Although ceramic fiber has better mechanical and fracture properties, ceramic fiber is not easy to disperse uniformly in silica sol slurry; excessive ceramic fiber will lead to the formation of a fiber beam. The creation of a fiber beam reduces the combination area of fiber and shell; the continuity of fiber and shell is seriously affected; and the combination between fiber and matrix becomes loose, so the fiber beam will reduce the enhancement effect of ceramic fiber on the shell. Additionally, the green bending strength of shell began to decrease. The good adhesion of fiber and colloidal silica shell is the key point to determine the enhancement effect of fiber-reinforced shell. Ceramic and nylon fibers increase green bending strength and prevent shell cracking when fiber disperses evenly in investment casting shell.

The green fracture morphology of fiber reinforced investment casting shell is shown in Figure 5. It is obvious that the distribution of ceramic and nylon fibers in the shell are uniform, which shows good mixing between fibers and slurry. The cross section of shell is irregular due to refractory material adhered to each other by fiber. Ceramic and nylon fiber distributes inside the investment casting shell; when shell is pulled apart, ceramic and nylon fibers will consume energy due to the combined interface being destroyed when fiber is separated from shell. Therefore, the strength of investment casting shell is improved. However, the strengthening effect will be affected by fiber features, such as type, content, distribution, and so on. As shown in Figure 5a, compared with ceramic fiber, nylon fiber is more uniform and has better compatibility with silica sol slurry, which leads to a good combination between fiber and silica sol slurry. The liquidity of composite fiber slurry is low and shell making will be easy to process; nylon fiber distributes the inside of the shell in a type of monofilament. The green bending strength of shell with nylon fiber is improved obviously, and higher than a shell with ceramic fiber. For ceramic fiber, as shown in Figure 5b, although mechanical performance is better than nylon fiber, ceramic fiber is more likely to reunite and forms fiber bundles inside the shell. The existence of fiber bundles will reduce shell performance.

### 3.3. Effect of Fiber Content on High Strength of Investment Casting Shell

The changing curves of high strength of shell after roasting with increasing ceramic and nylon fiber content is shown in Figure 6. It can be seen that the high strength of investment casting shell firstly increases then decreases with increase of fiber content. When fiber content increases from 0% to 0.3%, high strength of shell increases for both ceramic and nylon-fiber enhanced-shells, but the enhancement extent of ceramic fiber is better than nylon fiber; it is opposite to green bending strength, because most nylon fiber is burnt off. High strength of shell improvements were very limited; when ceramic and nylon fiber continues to increase and fiber content is more than 0.3%, the high strength of nylon-fiber enhanced-shell shows a sharp decline. At that point, the high strength of ceramic enhanced-shell continues to increase. When ceramic fiber content is 0.6%, high strength of that shell achieves the highest value—5.08 MPa; when the fiber content is bigger than 0.6%, the high strength of ceramic enhanced-shell begins to decrease. In general, the effect of fiber content on high and green bending strength is basically same and firstly increases then decreases, but the difference is the turning point. When fiber content is 0.6%, green bending strength of ceramic and nylon enhanced shell is the best; but for high strength, fiber content of ceramic and nylon is respectively 0.6% and 0.3% when high strength of shell achieves the best respective values.

The fracture morphology of fiber reinforced investment casting shell after roasting is shown in Figure 7. It can be seen that fracture morphology of nylon and ceramic enhanced-shell is rough. This is because silica sol slurry and refractory materials closely unify together after roasting. The high strength of the shell is significantly bigger than its green strength, and the fracture of ceramic enhanced-shell is much rougher than nylon enhanced shell. As shown in Figure 7a, nylon fiber will burn out after roasting because of a lower melting point. Thus, many micro pores are left inside shell; the existence of micro pores will reduce loading areas of shell. When the shell fractures, micro pores will split the shell using the internal defects. The high strength of the nylon-fiber enhanced-shell, reduces when nylon fiber content is higher. The effect of fragmentation is more obvious; the high strength of nylon-fiber enhanced-shell will continue to decrease. For a ceramic enhanced-shell after roasting, as shown in Figure 7b, because ceramic fiber is inorganic fiber and has some good properties, such as high mechanism performance, low heat conductivity, chemical stability and heat stability [32,33,34], ceramic fiber still remains inside the shell. Ceramic fiber and shell are closely connected to the whole. High strength of shell is improved due to ceramic fiber enhanced affect. When ceramic fiber content increases, high strength of shell continues increasing; when fiber content is too much, ceramic fiber reunites together and forms fiber bunch. That kind of fiber bunch will reduce the enhanced affect, and the high strength of the shell will decrease.

### 3.4. Distribution Morphology of Fiber Inside Investment Casting Shell

The fracture morphology of investment casting shell reflects characteristics of organization systematically; it will help to enhance the effects of fiber to shell. The fracture morphology of ceramic enhanced-shell with different fiber content is shown in Figure 8. These investment casting shells are not roasting. It can be seen from Figure 8a, shell fracture is smooth when fiber content is 0%; there is no especially obvious concave and convex shape. From Figure 8b–f, the number of fibers inside the shell increases gradually in the same area; when investment shells fracture, the fracture surface presents an uneven cross section. This is mainly because that fiber and refractory material tightly connect with each other, and fiber distributes in the cross structure inside the shell. The strength of investment casting shell is then significantly improved. It can be seen from Figure 8b–e, when fiber content is less than 0.6%, fiber composite silica sol slurry has good performance of coating; fiber is uniformly distributed in the shell, which proves that the dispersion of fiber in silica sol slurry is very even. Some fiber is pulled out of bonds with some slurry and refractory material. That indicates that when fiber enhanced shell under the action of external force, fiber will consume some energy because the bonding interface between fiber and shell is broken; this will delay the fracture of shell; therefore, the strength of shell will have a corresponding increase. When fiber content is 0.75% (fracture surface are shown in Figure 8f), fiber is unevenly distributed inside shell; some fiber presents slight bending, and some fiber bundles form, in serious cases, due to too much fiber. When fiber content is too high, the dispersion of fiber in silica sol slurry becomes worse which reduces contact surface between fiber and silica sol slurry. Fiber is prone to aggregate in the process of slurry making, so as fiber content continues to increase, the agglomeration of fiber aggravates, this the coating performance of silica sol slurry will become worse. The area of fiber bundles inside shell becomes loose. When the shell is damaged, fiber bundles will be pulled out; some refractory materials are pulled out also because of their interconnection between fiber bundles and shell, so bundles will improve shell strength, but the enhanced affect will decrease; meanwhile, fiber bundles will dissever strength of enhanced shell. When the negative effect is bigger than the positive effect, the strength of shell significantly decreases.

## 4. Conclusions

Ceramic and nylon fibers at different contents were mixed into silica sol slurry to reinforce shell. The effects of ceramic and nylon fibers on the liquidity of silica sol slurry and bending strength of investment casting shell were investigated. Results obtained in this study are summarized as follows:(1)The movement viscosity of silica sol slurries were all enhanced with an increase in ceramic and nylon fiber content; the movement viscosity of the ceramic fiber slurry was bigger than nylon fiber slurry, and the movement viscosity of the nylon fiber slurry was close to being a linear one;(2)The green and high strength of shell first increases, then decreases with the increases of fiber content. When the fiber content increased from 0% to 0.75%, when the fiber content was 0.6%, shell green bending strength reached maximum in both; the green strength of ceramic and nylon additions achieve 3.84 MPa and 3.56 MPa, respectively. The high strength of ceramic addition achieved its biggest at 5.08 MPa, higher than without the fiber. The high strength of nylon was biggest when fiber content is 0.3%;(3)The nylon fiber distribution was relatively uniform in shell matrix; the exposed nylon fiber of outside shell was straight, with no apparently broken trails. The poor compatibility of ceramic fiber caused the weak dispersion of the ceramic fiber in the matrix, and left holes as the part of fiber bundle was uprooted.

## Figures and Tables

**Figure 1 materials-12-02788-f001:**
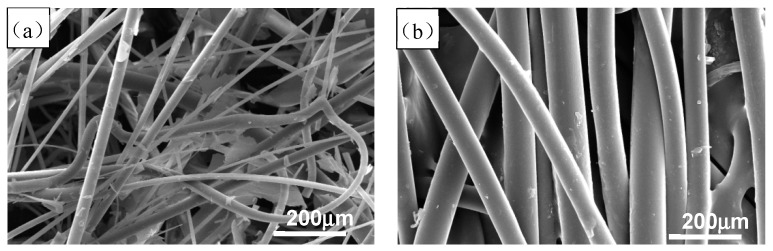
SEM image of nylon and ceramic fiber: (**a**) Ceramic fiber; (**b**) nylon fiber.

**Figure 2 materials-12-02788-f002:**
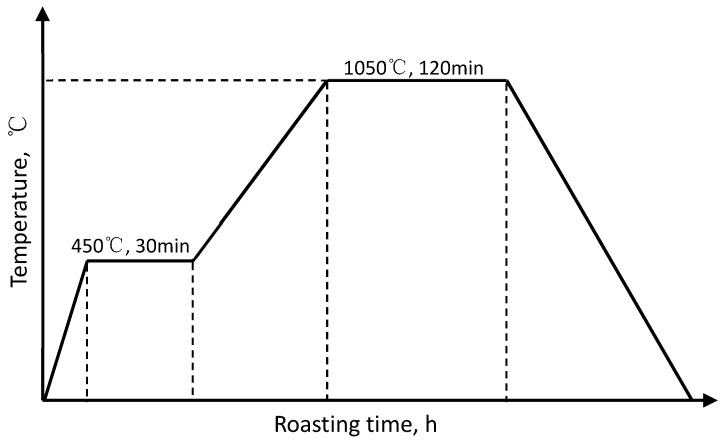
The process of shell roasting.

**Figure 3 materials-12-02788-f003:**
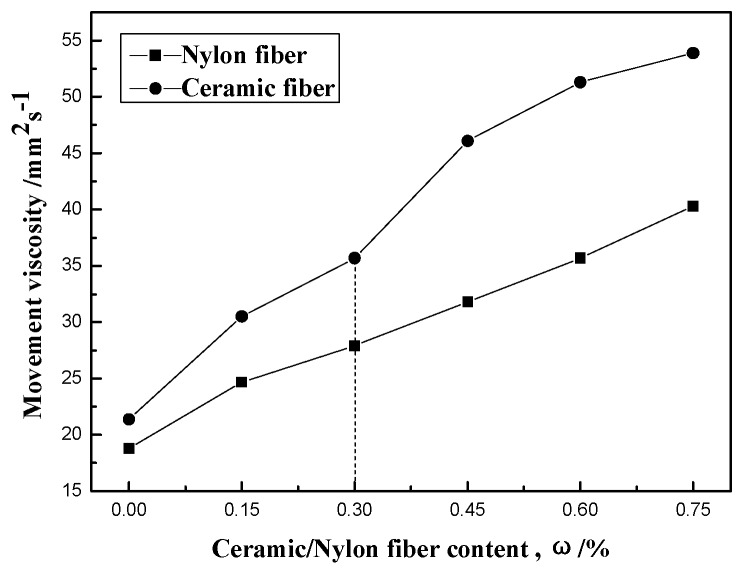
Influence of fiber content on the movement viscosity of composite silica sol slurry.

**Figure 4 materials-12-02788-f004:**
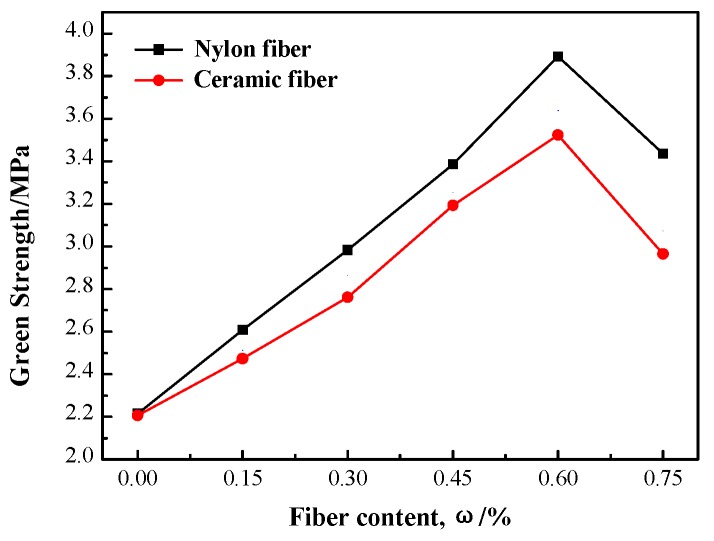
Effect of fiber content on the wet strength of investment casting shell.

**Figure 5 materials-12-02788-f005:**
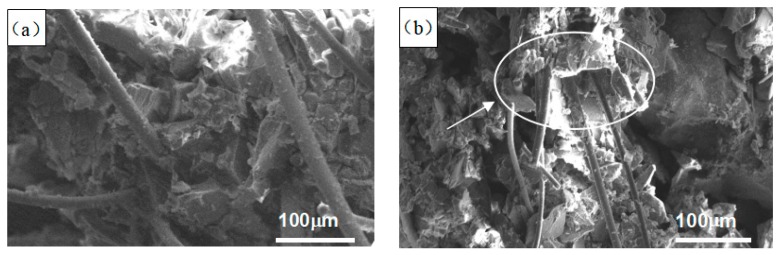
SEM image of green fracture surface of fiber reinforced investment casting shell: (**a**) 0.6 wt.% nylon fiber; (**b**) 0.6 wt.% ceramic fiber.

**Figure 6 materials-12-02788-f006:**
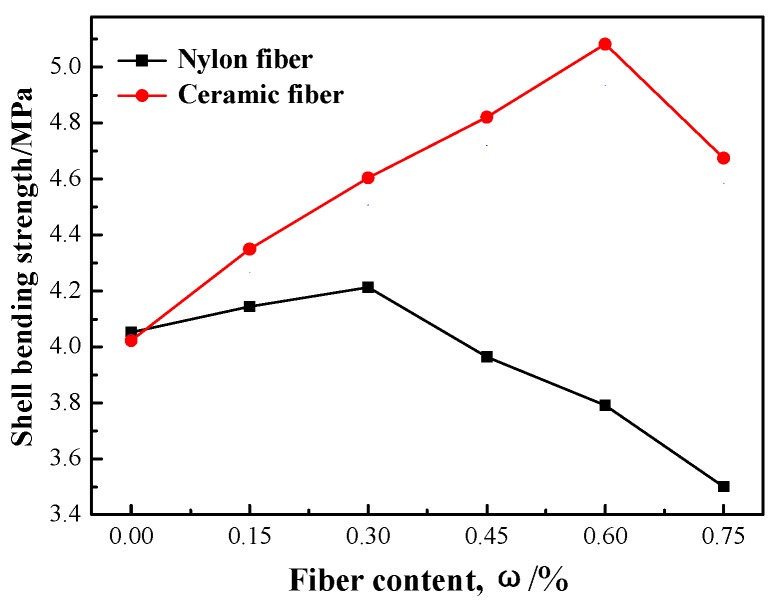
Influence of fiber content on the strength of shell roasting.

**Figure 7 materials-12-02788-f007:**
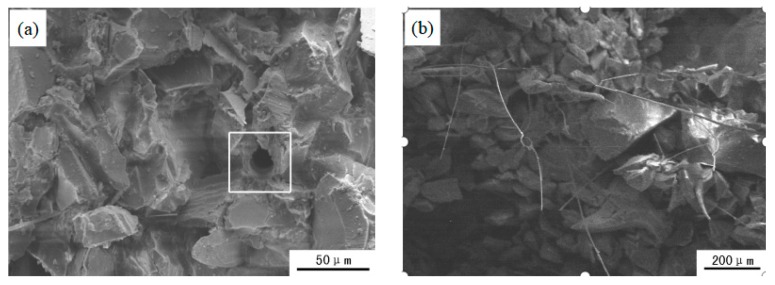
SEM image of fracture surface of investment casting shell after roasting: (**a**) nylon-fiber enhanced-shell; (**b**) ceramic enhanced-shell.

**Figure 8 materials-12-02788-f008:**
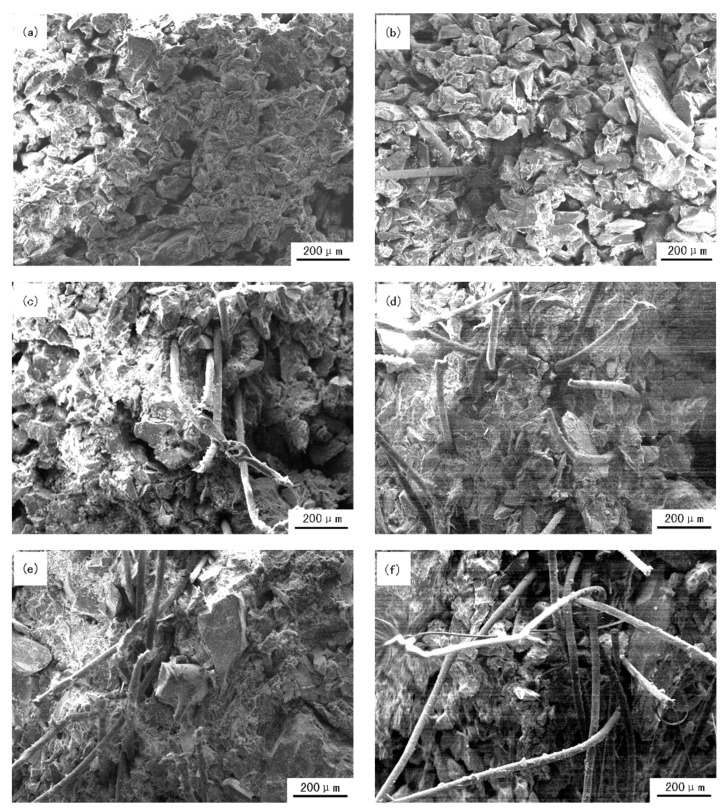
The distribution morphology of silica sol slurry shell with different fiber contents: (**a**) 0%; (**b**) 0.15%; (**c**) 0.3%; (**d**) 0.45%; (**e**) 0.6%; (**f**) 0.75%.

**Table 1 materials-12-02788-t001:** Main performance indicators of fibers.

Fibers Type	Length/mm	Diameter/μm	Tensile Strength/GPa	Elastic Coefficient/GPa	Density/(g. cm^−3^)	Melting Point/°C
Ceramic fiber	4~7	4~8	4	290	1.85	1800
Nylon fiber	4	9~13	0.9	5.17	1.16	224

**Table 2 materials-12-02788-t002:** Materials and process for the preparation of shell samples.

Coating Layer	Refractory Powder (Mesh)	Sanding Materials (Mesh)	Drying Time at Room Temperature (h)
Primary layer (1)	White corundum powder (320)	White corundum sand (100)	8
Transition layer (2)	White corundum powder (320)	Mullite sand (60)	12
Backup layer (3–5)	White corundum powder (320/100)	Mullite sand (46/20)	12
seal layer (6)	White corundum powder (320/100)	—	24

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
