# Peer review of "Effect of Ceramic and Nylon Fiber Content on Composite Silica Sol Slurry Properties and Bending Strength of Investment Casting Shell"

_materials, 2019, doi:10.3390/ma12172788_

Round 1

Reviewer 1 Report

Dear authors 

My comments are given in the attached file.

There are some typing mistakes, so please correct them.  Try to improve the introduction (add some references)

However, the manuscript is written well, so it can be accepted after these minor revisions. 

Regards

Author Response

Response to Reviewer 1 Comments

Point 1: Lines 10 to 12.Combine two sentences into one sentence

Response 1: We have merged the corresponding parts of the text into one sentence, and the detailed revision is shown in the red part of the 10-12 lines in the text.

Point 2: Space...typing mistakes...check in the whole text.

Response 2: We have checked the format of the full text and modified it.

Point 3: line 50,56,58. Add preferences.

Response 3: We have added references in the corresponding sections of the text. For detailed revisions, see the red, 50, 56, and 58 lines.

Point 4:  For LND-1,What this abravision means? Give the type of viscometer.

Response 4: The LND-1 is the viscometer type.

Point 5: Line 86.Change the corresponding part to italic.

Response 5:We have made changes.

Point 6: Dry time At which temperature? Room temperature?

Response 6: Yes,it is.We added the drying temperature to room temperature after the drying time.The modification is shown in the red part of Table 2.

Reviewer 2 Report

The manuscript entitled “Effect of ceramic and nylon fiber content on composite silica sol slurry properties and bending strength of investment casting shell” focused on the material characterization of composite material.

The manuscript is organized in an appropriate way, however a lot of work needs to be done before the publication process. First of all, there are a lot of typos and editing misstates. In many places of the manuscript, there is a lack of space between words and commas or dots. Level of English must be improved as well. The aim of the work needs to be clearly presented and the necessity of the research referring to the state of art should be presented.  

The authors should check that the presented value in Table 1 is not mixed, for elastic coefficient. The experimental part needs to be more precise with all of the necessary details.

Line 77 – it was mentioned, “Firstly, the surface of fiber was modified with a silane coupling agent, which can improve the wettability between fibers and silica sol…” – did the authors study the wettability of the material? And what type of silane coupling agent was used? All details about used chemicals must be presented.

Line 79 – there procedure of mixing is mentioned – what was the time and mixing conditions (rpm? Time? Temperature?)

Lines 99-100 – the analyzed samples by SEM were sputtered by conductive layer before SEM imaging? Please specified.

The plot axes should rather start from zero and the broken axis should be added.

The SEM images are not good quality, there are very dark and it difficult to follow, particularly Fig.8.

The obtained data are not supported by the literature. It should be compared and discussed.

Author Response

Response to Reviewer 2 Comments

Point 1: The manuscript is organized in an appropriate way, however a lot of work needs to be done before the publication process. First of all, there are a lot of typos and editing misstates. In many places of the manuscript, there is a lack of space between words and commas or dots. Level of English must be improved as well. The aim of the work needs to be clearly presented and the necessity of the research referring to the state of art should be presented.

Response 1:We have modified the format of the full text and modified some of the wrong sentences and words.We also elaborated on the purpose of the article.

Point 2:The authors should check that the presented value in Table 1 is not mixed, for elastic coefficient. 

Response 2: We checked the elasticity coefficient in Table 1 and confirmed it accordingly.

Point 3: Line 77 – it was mentioned, “Firstly, the surface of fiber was modified with a silane coupling agent, which can improve the wettability between fibers and silica sol…” – did the authors study the wettability of the material? And what type of silane coupling agent was used? All details about used chemicals must be presented.

Response 3:Our previous work investigated the wettability of materials, so we used silane coupling agent to increase the wettability of the material.The silane coupling agent we used was model CG-221 and was modified accordingly in the text.

Point 4: Line 79 – there procedure of mixing is mentioned – what was the time and mixing conditions (rpm? Time? Temperature?)

Response 4:The stirring time is 12h, the stirring temperature is room temperature, and the stirring speed is 120R/MIN. We have made corresponding modifications in the text.

Point 5: Lines 99-100 – the analyzed samples by SEM were sputtered by conductive layer before SEM imaging? Please specified.

Response 5: The analyzed samples by SEM were sputtered by conductive layer before SEM imaging. Sputtering distance of 50 mm, sputtering current of 10mA, sputtering time of 50 s.We have made corresponding changes in the text.

Point 6:The plot axes should rather start from zero and the broken axis should be added.

Response 6: After looking at the question raised by the reviewer, we found a lot of literature and found that the format of the graphs of many documents is consistent with ours, and we think that this setting can make readers understand the changes of our data better, so I did not modify it carefully. The teacher still thinks we need to modify, please let me know, thank you.

Point 7: The SEM images are not good quality, there are very dark and it difficult to follow, particularly Fig.8.

Response 7:  We have adjusted and replaced the image accordingly.

Point 8: The obtained data are not supported by the literature. It should be compared and discussed.

Response 8:We made a few modifications in the micro-organization analysis of Figure 8. We believe that the analysis can explain the problem. If the reviewer still feels that there is a problem, please let us know, thank you.

Reviewer 3 Report

Dear Editor,

Dear Authors,

the paper by Huang et al. entitled “Effect of ceramic and nylon fiber content on composite silica sol slurry properties and bending strength of investment casting shell” comprehensively elucidates the effect of these composites. The most important properties of the materials were investigated. The subject fits the journal scope and, in my opinion, is interesting for the scientific community. The introduction part brings the reader perfectly into the field.

Conclusions are clear and schematic.

In my opinion the MS deserves publication.

I have just some considerations.

The Abstract is too long and too descriptive. It shall be shortened to general information regarding the measurements undertaken. Please add proper and important references on investment casting shell considering also papers published in 2019 Please add an overall conclusion phrase to the <<Conclusions>> section. Some Typos were detected, just for instance:

Line 166: change “is destroy” with “is destroyed”

Line 206: consider to change “exist” for “existence”

Line 223: change “…structure and was systemic observed…” in “systematically”

change: “it will be help to understand” in “it will help to”.

Line 125: change “lead” in “leads”.

Based on the above arguments, I recommend the paper for publication after minor revision.

Author Response

Response to Reviewer 3 Comments

Point 1: The Abstract is too long and too descriptive. It shall be shortened to general information regarding the measurements undertaken. Please add proper and important references on investment casting shell considering also papers published in 2019 Please add an overall conclusion phrase to the <> section. 

Response 1:  We have made corresponding changes to the abstract section and added 2019 references to the introduction. We have added an overall conclusion phrase to the conclusion section.

Point 2: Some Typos were detected, just for instance:

Line 166: change “is destroy” with “is destroyed”

Line 206: consider to change “exist” for “existence”

Line 223: change “…structure and was systemic observed…” in “systematically”

change: “it will be help to understand” in “it will help to”.

Line 125: change “lead” in “leads”.

Response 2: All the places we mentioned were revised according to the comments of the reviewer.

Round 2

Reviewer 2 Report

The manuscript has been corrected. All unclear issues have been clarified. The manuscript is in the suitable form to Ben published.